# PyMC4: Exploiting Coroutines for Implementing a Probabilistic Programming Framework

**Max Kochurov**
Skoltech
Moscow, Russia
`maxim.v.kochurov@gmail.com`

**Colin Carroll**
Cambridge, Massachusetts
`colcarroll@gmail.com`

**Thomas Wiecki**
Quantopian Inc.
Düsseldorf, Germany
`thomas.wiecki@gmail.com`

**Junpeng Lao**
Google Inc.,
Zurich, Switzerland
`junpenglao@google.com`

## 1  Introduction

PyMC4[1] is an open-source probabilistic programming library whose goal is to give users access to cutting-edge algorithms in Bayesian statistical computing while being extensible enough to help researchers to implement novel algorithms. Like its predecessor PyMC3 [10] and the C++ library Stan [5], the aim of PyMC4 is to provide users with a high-level API to specify probabilistic models. Our focus is on Bayesian models where inference can be performed using (dynamic) Hamiltonian Monte Carlo and variational inference, both of which require automatic differentiation of user-defined joint density functions.

PyMC4 is built on top of TensorFlow [8] (TF) and the TensorFlow Probability (TFP) library [3], inheriting and extending common distributions, invertible transformations, and inference algorithms, all of which rely on TensorFlow for automatic differentiation and modern hardware binding. This allows PyMC4 to focus on intuitive syntax for model specification, model introspection, and automatic parameterization. PyMC4 uses coroutines [2] to dynamically control and update the program flow.

The previous version of PyMC (PyMC3) is built on top of Theano, which provides automatic differentiation and advanced linear algebra necessary to build a probabilistic programming framework. PyMC3 is tightly integrated with Theano[2]: it relies on the creation of a compiled (static) graph that represents the log density function and its gradient with respect to the free parameters of the model. Furthermore, Theano allows for convenient graph introspection and copying, on which PyMC3 relied heavily to generate a closure of the target density function that conditioned on the observed variable and/or part of the free parameters. In TensorFlow, graph manipulation is generally discouraged. Moreover, the new TF 2.0 API is a dynamic computational graph model, which makes tracing and editing of the computational graph even more challenging. Together, the lack of graph modification and the transition from static to dynamic graphs introduce trade-offs for the implementation of a probabilistic programming framework. Although dynamic computation allows for a more imperative programming style (similar to Python), a directed acyclic graph is no longer available for introspection and manipulation.

From an API point of view, dynamic graphs introduce a new level of abstraction for models. To make PyMC4 user-friendly, a design goal is that a model's specification should not change depending on whether it is used for inference, evaluation or debugging. However, the program execution order and

---

[1] `https://github.com/pymc-devs/pymc4`
[2] `https://docs.pymc.io/developer_guide.html`

Preprint. Under review.

the resulting computational graphs are different for these three purposes. In PyMC4, we solve this problem using Python coroutines as explained in more detail below.

## 2    Intercepting control flow with Python coroutines

With PEP342 [9], Python gained enhanced generators and coroutines. A coroutine call returns a value from the coroutine to the main program and suspends operation until the main program sends a value back to the coroutine which resumes it. This is crucial for writing PyMC4 on top of a dynamic graph; once a model is written as a coroutine, the execution engine may change the control flow of the model as needed.

More specifically, there are three contexts in which a model in PyMC4 can be run:

1. Generating samples from the prior predictive distribution (e.g., for diagnostics [4, 11]);
2. Generating a Python callable that represents the joint probability distribution conditioned on observed data for inference (sampling or optimization);
3. Generating samples from the posterior predictive distribution conditioned on the inference result.

Consider the following toy implementation of PyMC4:

```python
def control_flow(model, **observed):
    # A placeholder, to be replace with a sample or observed value.
    current_value = None
    while True:
        try:
            # Interact with the model via coroutine
            random_variable = model.send(current_value)
        except StopIteration as e:
            # return statement may appear in generator
            return e.args and e.args[0] or None
        else:
            current_value = observed.get(random_variable.name)
            if current_value is None:
                current_value = random_variable.sample()
            yield random_variable, current_value
```

In this program, the argument `model` could be the following, where `Normal` has a `name` attribute, and a `sample` method:

```python
def model():
    x = yield Normal("x", 0, 1)
    y = yield Normal("y", x, 1)
```

Prior predictive sampling is done by running the model forwards, without any observations:

```python
def prior_predictive(model):
    prior_samples = {}
    for rv, sample in control_flow(model):
        prior_samples[rv.name] = sample
    return prior_samples
```

The joint density function could be constructed using a Python closure, of which the output is a Python callable, with observations passed in to the proper nodes:

```python
def construct_log_pdf(model, **observed):
    def log_pdf(**vars):
        total = 0
        for rv, sample in control_flow(model, **vars, **observed):
            total += rv.log_pdf(sample)
        return total
    return log_pdf
```

Finally, we could sample from the posterior predictive distribution in a similar way as to the prior predictive distribution, conditioned on the inference result from above:

```python
def posterior_predictive(model, **posterior):
    conditional_samples = {}
    for rv, sample in control_flow(model, **posterior):
        conditional_samples[rv.name] = sample
    return conditional_samples
```

## 3 Program transformations

Implementing models as coroutines allows for manipulation of its run-time behaviour and replacement of execution code. As PyMC4 unwraps any nested control flow, it becomes easy to replace a random variable obtained from user code with its reparametrized submodel version.

```python
# inside def control_flow(model, **observed):
    # ...
    rv = model.send(sample)
    if need_reparametrize(rv):
        # Recursive call
        sample = yield from control_flow(reparametrize(rv),
                                         **observed)
    # ...
```

These transformations also allow us to automatically rewrite a model for easier sampling, e.g. replacing constrained variables with unconstrained ones to allow HMC sampling or ADVI [6].

In addition, we can also reparametrize whole (sub-)models [7] to get rid of ill conditioned or challenging posterior regions like funnels. For example, one may consider the following model reparametrization for a single normally distributed random variable.

```python
def reparametrize_normal(normal_dist):
    mu, sigma = normal_dist.mu, normal_dist.sigma
    eps = yield Normal(normal_dist.name + "_eps",
                       zeros_like(mu), ones_like(sigma))

    return mu + eps * sigma
```

In the above example, instead of sampling and computing the log probability in the original parameterization, an alternative reparameterization should be used. The transformations are implemented programmatically and may be applied on demand giving flexibility in exploring alternative model specifications with little effort.

## 4 Composing models through nesting

Bayesian models are often composed of connected submodels. For example, a horseshoe prior [1] is used like a single distribution but is usually expressed as three connected but individual distributions:

$$\beta \sim \mathcal{N}(0, \tau^2 \lambda^2)$$
$$\lambda \sim \mathcal{C}_+(0, 1)$$
$$\tau \sim \mathcal{C}_+(0, \tau_0).$$

In order to facilitate model reuse, PyMC4 supports model composition through recursive nesting:

```python
@model
def Horseshoe(tau0, n):
    tau = yield HalfCauchy("tau", 0, tau0)
    delta = yield HalfCauchy("delta", 0, 1, plate=n)
    beta = yield Normal("beta", 0, tau * delta)
    return beta
```

The `plate` argument specifies the number of independent realizations of the created distribution.

```python
@model
def model(observed):
    sparsity_prior = yield Horseshoe("sparsity_prior", 1,
                                     len(observed))
    # ...
```

This functionality abstracts submodels, composed of multiple probability distributions, to be treated like a single probability distribution in other parts of the model.

## 5 Acknowledgements

We would like to thank Rif Saurus, Christopher Suter and the TFP team for valuable discussions on the PyMC4 design as well as helping with technical questions. We would also like to thank George Ho, Ravin Kumar, Osvaldo Martin, Robert Goldman and Christopher Fonnesbeck for feedback and revisions on an earlier draft. Funding for developer meetings was provided by Google.

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
