# OpenReview forum: "PyMC4: Exploiting Coroutines for Implementing a Probabilistic Programming Framework"
_NeurIPS.cc/2019/Workshop/Program_Transformations — Program Transformations @NeurIPS2019 Poster_

### Official Review · AnonReviewer1 · 2019-09-28
**A new PyMC based on TensorFlow**

**Confidence:** 4
**Rating:** 8

**Review:**

This paper introduces the PyMC4 probabilistic programming library. PyMC4 is based on TensorFlow for automatic differentiation and hardware bindings, unlike the previous version, PyMC3, which was based on Theano. The authors report that moving the project to TensorFlow 2.0 with dynamic computation graphs necessitated a new design and new level of abstraction. The authors use Python’s coroutines feature to intercept control flow and implement this new design.

The paper is interesting, well-written, and easy to follow. I would be interested in learning more about the design challenges and how the authors settled on a coroutine-based scheme; in particular, I am curious whether they had any other alternative designs (without coroutines) that they ended up not using. I also like the fact that the paper presents little code boxes showing the implementation details of PyMC4’s internals.

Minor: in Section 4, the code refers to a “delta” value for one of the half-Cauchy distributions, but the equations above use the letter “lambda”.

---

### Official Review · AnonReviewer2 · 2019-09-29
**Needs discussion of limitations and a non-trivial example**

**Confidence:** 3
**Rating:** 6

**Review:**

This package seems sensible enough, but
 - It's not clear how much is gained on top of TF Probability,
 - I'd like to see a discussion of what sorts of priors can't be expressed this way (e.g. what about deterministic transforms?)
 - A discussion of how this conditioning approach scales
 - I personally think it's ugly that you need to pass around strings containing variable names.

---

### Decision · Program_Chairs · 2019-10-01

**Decision:**

Accept (Poster)

**Comment:**

This abstract introduces the new PyMC4 library. The reviewers thought this was a good contribution, although it raised a few questions (e.g., what gains are there compared to TF probability and why were certain design decisions made).